



# Title: A revisiting of early 18th century environmental data to identify Gulf of Lion properties before the industrial era

Short title: Gulf of Lion environmental properties in early '700.
Marina Locritani[1], Sara Garvani[1,2], Giuseppe Manzella[3], Giancarlo Tamburello[1], Antonio Guarnieri[4]
[1]Istituto Nazionale di Geofisica e Vulcanologia, Roma, 00143, Italy
[2]Università di Genova, Genova, 16126, Italy
[3]OceanHis, Torino, 10153, Italy
[4]Istituto Nazionale di Geofisica e Vulcanologia, Roma, 00143, Italy, now at DG MARE - European Commission,
Bruxelles, 1049, Belgium
*Correspondence to*: Marina Locritani (marina.locritani@ingv.it)
Keywords: historical oceanography, Marsili, Marsigli, Histoire Physique de la Mer, bathymetry, water specific gravity, density and
salinity, temperature, sea level, Gulf of Lion, Mediterranean Sea
**Abstract**
The work "*Histoire Physique de la mer*", authored by Luigi Ferdinando Marsili (or Marsigli) and published in 1725, is
one of the earliest texts detailing observations of the physical, biological, and bathymetric characteristics of the sea,
mainly concentrating on the Gulf of Lion in southern France's Mediterranean area. Nonetheless, understanding Marsili's
findings is difficult due to the application of non-standard measurement units and the imprecision of georeferencing data.
The MACMAP project (A Multidisciplinary Analysis of Climate Change Indicators in the Mediterranean and Polar
Regions), which is funded by the Istituto Nazionale di Geofisica e Vulcanologia (INGV), has involved a thorough
recalibration of Marsili's observations. This project focused on transforming water weight measurements obtained from
different locations in the Gulf of Lion from June 1806 to January 1807 into water density values. The sampling sites were
digitized, bathymetric profiles were reconstructed, and tide amplitudes were examined. The main objective is to make
this historical data available to compare with current measurements.
**1. Introduction**
The study of oceanography took a methodological turn with the 'recommendations' made by Murray and Hooke in 1666,
which emphasized the need for 'diligent' observations from 'masters of ships, pilots, and other fit individuals during their
sea voyages.' These 'recommendations' outlined various measures, along with the methodologies and technologies that
should be employed. A significant enhancement to the concept of 'diligent observations' at sea was introduced by Luigi
Ferdinando Marsili, a member of the Society who stressed the importance of a robust sampling strategy to analyze the
'physical nature of the sea'. Marsili's contributions to oceanography were largely recognized again during the 300th
anniversary of his birth, thanks primarily to Olson and Olson's article, "Luigi Ferdinando Marsigli, the lost father of
oceanography" (1958), as well as Deacon's book (1971).
Marsili led a dynamic life as a traveler, soldier, and scientist (see Appendix B for more details). His most renowned work,
*Osservazioni Interne al Bosforo Tracio* (1685), outlines the two-layer circulation model that later elucidated the flow into
and out of the Gibraltar Strait. Also significant is *Histoire Physique de la Mer* (1725), an extensively illustrated
publication offering invaluable historical and scientific insights on the Gulf of Lion (the digital version of this volume
can be accessed here: https://www.google.it/books/edition/Histoire_physique_de_la_mer/RpsSxRY4bRcC).
Marsili was affiliated with the Paris Académie Royale des Sciences and the Royal Society of London. He greatly admired
Robert Boyle but was critical of Boyle's work "De fundo Maris," which he addressed in the first chapter of the Histoire,
where he asserted the necessity of verifying observations without dismissing sailors' theories. To fully grasp the
innovative concepts in the Histoire, one must understand the author's discussions with other Académie des Sciences de
Paris members. Initially, Marsili believed that stony formations like coral and Madrepores were not true plants (contrary
to the views of Joseph Pitton de Tournefort and others), but rather 'drips' created by the sea's viscous material (Archives
Départementales de l'Héraut, Registre de la Société royale des sciences de Montpellier, Hérault D. 116; Memoires de la
Société Royale in Histoire de la Société royale des sciences établie à Montpellier, 1778), as noted by McConnell (1990).
Through his Observations, Marsili ultimately realized his assumption was mistaken. However, it's crucial to acknowledge
his investigative approach, grounded in the Baconian framework of science, which posits that understanding nature begins





with collecting and methodically examining facts. To assess coral growth and explore those 'drips' or 'coral milk,' he conducted physical measurements of various sea properties, including sea level, currents, and density.

Marsili's goals extended beyond simply determining whether corals were flowers or something different. He sought to create a comprehensive natural history of the sea, its seabed, wind effects, and fish characteristics. His ambition was to compile a significant treatise on the Earth's organic structure. A key component of this research involved studying lakes and sea structures, underpinned by the belief in a morphological and lithological connection between mountains and the seabed, all vital for his theoretical development. 'My intention [...] is to show the organic structure of the Earth' (first chapter of the Histoire, dedicated to the marine basin). Marsili explored the Languedoc mountains, but while this research remained unpublished, his investigations of the sea first culminated in the Brieve Ristretto (1711), an essay printed in Venice, and later in the *Histoire* of 1725.

This is not the first occasion to analyze Marsili's measurements compared to contemporary data. This comparison supports the accuracy and meticulousness of the Bolognese general's data collection methods. The seawater weight data gathered by Marsili in the Constantinople Channel, as detailed in *Osservazioni intorno al Bosforo Tracio,* were examined by Soffientino and Pilson (2005). Their analysis indicated that Marsili's findings were inconsistent with current data because they were treated as salinity data before being adjusted to align with the figures in Zupko's work (Zupko, 1981). In contrast, Pinardi et al. (2018) interpreted the Bosphorus seawater weight data as density data, finding it consistent with modern measurements.

This paper compares physical data— seawater weight, bathymetric profiles, and sea level variations—found in *Histoire Physique de la Mer* to current measurements. This research is part of the MACMAP project (A Multidisciplinary Analysis of Climate Change Indicators in the Mediterranean and Polar Regions), which receives funding from the Istituto Nazionale di Geofisica e Vulcanologia (INGV).

Data on seawater weight from the *Histoire Physique de la Mer* were transformed into density values using details from the book's preface (Marsili, 1725). A specific, unnumbered page in the preface states: "*J'ai pesé les Corps solides avec la Balance ordinaire, mais très-exacte, me servant de la livre composée de douze onces, l'once de huit dragmes, & la de la dragme de soixante grains; & les fluides avec l'Areometre de verre, de la forme, grandeur, & poids que l'on trouvera décrits*." The locations of the sampling stations were determined using maps and georeferenced within a geographic system, facilitating a comparison between Marsili's data and contemporary measurements from the exact locations. Marsili's water samples represent one of the earliest examples of modern field sampling. He employs a methodical approach that aligns with the reproducibility principle of measurements, enabling comparisons between seventeenth-century collected data and current datasets (Pinardi et al., 2018).

The bathymetric profiles illustrated in Table III of *Histoire* have been digitized and contrasted with the transects presented in Table II, which have been georeferenced for this study. The findings have been analyzed alongside "modern" data. Variations in sea level listed in Tables X and XI of *Histoire* have also been examined and compared with "modern" data.

## 2. Material and methods

### 2.1. The Histoire Physique de la Mer

Luigi Ferdinando *Marsili's "Historie physique de la mer,"* published in 1725, details the physical, biological, and bathymetric attributes of the Provençal coastal region up to the shelf break. Across all his works, from *"Opus Danubialis"* to his *"Histoire Physique de la mer*," Marsili embraced a humble yet ambitious "Baconian" approach to science. During his tenure as a military engineer, he undertook comprehensive scientific studies that equipped him with a systematic method, which became particularly beneficial after settling on France's southern coast in 1706. While engaged in maritime research, he authored a treatise on the seabed and its waters, dedicating a significant portion to corals and other lithophytes erroneously identified as plants.

The *Histoire* was initially printed across 173 pages, featuring 40 plates depicting animals, minerals, fossils, and plants, presented as 12 fold-out plates containing large-scale data, maps of the Gulf of Lion, and coastal profiles. Luigi Ferdinando Marsili employed a strong methodology for obtaining quantitative, well-organized in situ seawater density measurements, including precise time and location specifications. Marsili's primary instruments for data collection consisted of a thermometer, an areometer to assess water weight, a sampler for surface and depth water collection, a "depth gauge," and a metered pole for evaluating sea level variations.





### 2.1.1. Bathymetry and sea level

In the first chapter, *Du Bassin* illustrates the bathymetry using graphs that detail the continental shelf's edge. Marsili measured the Gulf of Lion's depth at 14 points, which he represented on a map. These findings allowed him to outline both the seabed and coastal profiles. This volume contains the initial map of the Gulf of Lion, showcasing the bathymetric slope that separates the continental shelf from the abyssal plain (Table I, page 3, titled *Carte du Golfe del Lion entre le Cap Sisie en Provence et le Cap de Quiers en Roussillon*). Another map depicts the coastline from Cap Canaille to the Croisette and the nearby islands within Cassis' territory in Provence (Carte Particuliere de la Coste, Table II, page 4). Bathymetric profiles can be found in Table III, page 4 (Profils ou Coupes du Bassin de la Mer), Table IV, page 4 (*Profils ou Coupes du Bassin de la Mer sur la Coste de Provence*), and Table V, page 7 (Porte Miou). Neither map features a Coordinate Reference System and they are based on earlier maps. In the volume's preface, Marsili credits the prior research that informed his reconstruction of the Gulf of Lion's bathymetry, including De Basville's study of the Languedoc coast and De Chazelles' map of Provence and Roussillon, who was also an Engineer des Galères and a member of the Paris Academy of Sciences and a hydrography professor in Marseilles.

Marsili determined the depth using a traditional method involving a weight on a graduated rope. He heavily relied on measurements provided by fishermen, who would say 'the abyss has no bottom' (l'Abîme n'a point de fond) when depths were beyond their measuring lines. Marsili considered this saying inaccurate, as he aligned with the prevailing thought of his time that sea bathymetry reflected terrestrial altimetry.

To gauge changes in sea level near a coastal section and the Cassis Sea basin, Marsili employed a metered pole. From January 4 to April 9, 1707, he recorded the water's elevations and depressions at various times. He details his study of sea level variation conducted at Cassis port in the third chapter of *Histoire*.

### 2.1.2. Physical characteristics

The Histoire details temperature and salinity, featuring illustrations of the instruments used and data collection tables. Information on temperature can be found in Table VI, on page 16, titled *"des Experiences fautes avec le Thermometre dans la mer à differentes profondeurs."* The sea water temperature was measured using a Florence thermometer, as suggested by Cotte (1774). Marsili likely utilized a three-hook wine-spirit thermometer, similar to those detailed by Camuffo (2020). Measurements were taken at the sea surface and various depths by securing the thermometer to a rope weighted appropriately. Marsili's thermometer was attached to a wooden board, featuring a double scale that is numbered in reverse: 1-55 and 55-1. The thermometer consisted of worked glass supported by three iron wires attached to the board. Two horizontal hooks were standard, located above and below, while an unusual third vertical hook at the top prevented the thermometer from sliding down (Cotte 1774, Camuffo, 2020). Cotte was unable to convert the temperature readings from Marsili's scale to Reaumur units, and sadly, Marsili's thermometer was lost in a maritime conflict in 1707 with an enemy brig or pirate vessel. Consequently, the thermometer's scale remains unknown today, and we lack information about the duration for which the instrument was submerged, despite Marsili's meticulous record of the time immersion.

In Marsili's time, the concept of water density was understood in terms of sea gravity. As noted by Manzella et al. (2021), measurements of gravity and salinity were conducted using a known-sized vial with a narrow neck or a graduated glass tube. Gravity was assessed by weighing the water, while salinity was calculated based on the weight of the residue left after evaporation.

The water weight was gauged using the hydrostatic ampoule (also known as the hydrostatic carafe or areometer, see Locritani & Garvani, 2024 and Appendix B), which consists of a sealed glass sphere with a tapering neck (Montanari, 1696). Viviani, a student of Galileo, described the measurement process: the ampoules needed to include enough lead flakes so they would float in the liquid being assessed. Additional known-weight rings were placed on the neck until the ampoule was submerged. The weight of these rings that caused the ampoule to sink equaled the weight of the measured water. For further details, see Pinardi et al. (2018). Sea water samples were collected using a wooden container sealed with a valve. Marsili first weighed the sea water onboard the vessel and then used a balance later in a lab. The reference water was sourced from a well near Marsili's lab in Cassis. Marsili gathered water samples between Cap Canaille and Cap Croisette, which he subsequently analyzed in his lab using a thermometer and the hydrostatic ampoule or balance. The hydrostatic method involved measuring an object's weight by submerging it in a liquid of known weight. The second chapter includes tables and figures that present data on the weight of salty and fresh waters, a description of the areometer or hydrostatic ampoule, and the salt concentration in the water samples.





### 2.1.3. Vegetation

The fourth chapter of the *Histoire*, titled *De la vegetation des plantes*, is the largest section. It features 40 copper-engraved illustrations by Matthys Pool (1676-1740), depicting animals, minerals, fossils, and plants, all numbered with corresponding references in the text. This section includes many plates that portray corals, which Marsili classified as "plants." Each plate indicates the location where the corals were discovered and describes how they were gathered by fishermen, along with the local names or those assigned by Marsili, as Linnaeus' nomenclature had not yet been adopted.

## 3. Data conversion

### 3.1. Bathymetry

The initial step in extracting historically significant information from historical-geographical maps involves defining a set of immutable and identifiable ground control points (GCPs) for georeferencing these maps in QGIS and then digitizing the sampling points. The scale of the Marsili maps is expressed in "toises" or "teses", which correspond to 1.2 brasses marine (fathoms), an ancient unit of length roughly equal to 0.32 feet or 1.95 meters, and according to Angelo Martini's metrology manual, 1 meter is approximately 0.001 mile marine (Martini, 1881). The direction and length of the transect propagation is measured in miles, nearly equivalent to nautical miles; specifically, 1 marine mile equals 1851 meters, as noted by Martini (1881, p. 466). In the initial phase, distances from the Marsili maps were obtained using Adobe Illustrator 2022 and compared with those in contemporary maps (Google Satellite in QGIS 3.22). The mean, standard deviation, and root mean square for distances between pairs of GCPs from both the Marsili and Google Satellite maps were calculated in QGIS 3.22. Following recommendations by Bitterer (2006) and Hvizdák (2023), a statistically adequate sample requires GCPs greater than 30, which poses challenges with historical maps. To align the ancient and modern maps more closely, a restricted area near Marseille (from Toulon to Port de Bouc) was selected, resulting in 8 GCPs being recorded and an additional 4 GCPs zoomed in on the Cassis area. The second map was entirely georeferenced using these 8 GCPs. The coordinates of the GCPs are detailed in Table 1. Subsequently, the maps were georeferenced in QGIS 3.22 applying "Thin Plate Spline (TPS)" interpolation and the "Nearest Neighbour" sampling method.

| Cassis form Pl II pag 4 | | |
|---|---|---|
| GCP | Latitude [°N] | Longitude [°E] |
| 1 | 43.184196° | 5.563402° |
| 2 | 43.207667° | 5.368603° |
| 3 | 43.212021° | 5.539110° |
| 4 | 43.211178° | 5.337976° |
| 5 | 43.175865° | 5.382420° |
| 6 | 43.203079° | 5.511857° |
| 7 | 43.204568° | 5.426200° |
| 8 | 43.203280° | 5.452809° |
| **Marseille from Pl I pag 3** | | |
| GCP | Latitude [°N] | Longitude [°E] |
| 1 | 43.278345° | 4.890243° |
| 2 | 43.282043° | 5.346703° |
| 3 | 43.279782° | 5.325148° |
| 4 | 43.393690° | 4.985789° |
| 5 | 43.045905° | 5.859005° |
| 6 | 43.213436° | 5.337643° |
| 7 | 43.341584° | 5.265206° |
| 8 | 42.880143° | 5.308412° |
| **Cassis from from Pl I pag 3** | | |
| GCP | Latitude [°N] | Longitude [°E] |
| 1 | 43.214906° | 5.336222° |
| 2 | 43.160508° | 5.607515° |
| 3 | 43.173813° | 5.398702° |
| 4 | 43.209136° | 5.539286° |

**Table 1: GCP used to georeferenced the Marsili maps (Gulf of Lion map - Pl. I, page 3 in the Marseille coastal area and the Cassis coastal area and Cassis map - Pl. II, page 4) in QGIS 3.2.**

Additionally, the EMODnet_satellite_coastline_MSL has been superimposed onto the georeferenced historical maps. Upon completing this georeferentiation, we could extrapolate the coordinates of the small urns and the positions of the transects.

The bathymetric profiles were digitized using WebPlotDigitizer (Rohatgi, 2017). The profile length in nautical miles (nmi) is indicated in the legend at the upper-right corner of the original figure. This facilitated the conversion of the horizontal distance from pixels to nautical miles. The maps display lines representing transects along which Marsili conducted bathymetric measurements. Descriptions in the *Histoire* tables provide starting points, total transect lengths, and depth measurements in fathoms in a few locations (*brasses;*



1 ftm is 1.624197 m, Martini, 1881); this data was used to convert the vertical distance from pixels to meters.
Some profiles lacked depth information; therefore, they were digitized in adimensional units (zero for sea level
and -1 for the profile's maximum depth). To compare with current bathymetry, the spatial coordinates of the
georeferenced transects from the maps were utilized to extrapolate bathymetric values from EmodNET
(https://emodnet.ec.europa.eu/geoviewer/) and Gebco 2023 (https://download.gebco.net/) data in QGIS 3.22.

### 3.2. Sea Level in Cassis

The Marsili sea level measurements were conducted not in a well but likely in a protected coastal area using a 68 pouces
rod, referred to as an inch rod by Marsili (1 French inch is roughly 2.7 cm, according to Martini, 1881). A timing issue
exists; the measurements lacked precise timestamps (such as sunrise, noon, sunset around 9 pm, and midnight). For the
analysis, measurements from sunrise to sunset were assumed to be recorded at roughly 6:45 am, noon, 6:45 pm, 9 pm,
and midnight. It is also assumed that each time point included two measurements (maximum and minimum) to mitigate
wave effects through averaging.
A data series was created using Marsili's data from January 5 to April 10, 1707. The minimum and maximum values were
averaged at each time interval, and then the mean was subtracted. The resultant series exhibited significant noise, leading
to the application of multiple filters to identify key signals. A primary low-pass filter was introduced to smooth the time
series. Notably, periodicities of about 5-7 days and 13-14 days were observed, which may relate to atmospheric influences
(Esposito and Manzella, 1982).

### 3.3. Density

Water weight measurements were recorded in ounces, drachmas, and grains. Maps marked the water sampling sites using
the amphora symbol. To convert the weight to Kg/m³, it was assumed that 1 pound corresponds to 12 ounces, 1 ounce to
8 drachmas, and 1 drachma to 60 grains (Marsili, 1725). The transformation formula used to convert weight to density is
$\rho=1000*P/Pr$, where $\rho$ represents the density of water in Kg/m³; 1000 is the density of distilled water in Kg/m³; P is the
weight of seawater in grains as measured by Marsili; and Pr is the weight of reference water in grains from Marsili's
measurements. The minimum potential error in the measurements was assessed by considering values below 1000 Kg/m³
for freshwater (such as rivers, fountains, and wells). This data has been compared with the water density measurements
from SeaDataNet covering the same sites from 1990 to 2018.

## 4. Results and Discussion

### 4.1. Bathymetry: historical map analysis

Marsili's maps include scales that indicate distances of 10000, 5000, 1000, 500, or 100 "teses". Due to the numerous
variations, it was necessary to evaluate the consistency within each map and then compute the distances. Table 1 lists 20
Ground Control Points (GCPs) selected for distance calculations from the Marsili maps, showing the relative distances
measured between pairs of GCPs in both maps, along with the differences in lengths (errors). Table 2 presents the Root
Mean Square Error (RMSE), as well as the means and standard deviations. The RMSE for the Gulf du Lion map is 6087
m, while for the Cassis map, it is 548 m. The mean distances are 2912 ± 6057 m for the Gulf du Lion map and 324 ± 472
m for the Cassis map. The errors are uneven, evidenced by the standard deviations exceeding the mean values. Improved
results are achieved by remapping the coastline and bathymetric lines for more localized areas (refer to Figure A1 in
Appendix A). This adjustment reduces the error, resulting in a comparison with EMODnet coastlines and bathymetric
lines showing an error of 631.38 ± 559.72 m for the Gulf of Lion and 163.83 ± 157.33 m for the Cassis area.




| Site name | Latitude start [°N] | Longitude start [°E] | Latitude end [°N] | Longitude end [°E] | Lenght in Marsili map [m] | Lenght in Google Satellite [m] | Error beetween new - old |
|---|---|---|---|---|---|---|---|
| **Map of Cassis coastal area** | | | | | | | |
| Ile Riou | 43,179382° | 5,371768° | 43,173686° | 5,398688° | 2006,10 | 2274,00 | 267,90 |
| Ile maire | 43,212390° | 5,328571° | 43,209194° | 5,340005° | 715,52 | 996,71 | 281,19 |
| Cap Croisette-Cassis | 43,214517° | 5,327929° | 43,214658° | 5,537063° | 14872,38 | 16333,00 | 1460,62 |
| Calanque de Port Pin | 43,197710° | 5,509915° | 43,203919° | 5,510848° | 667,24 | 695,00 | 27,76 |
| Cap Croisette | 43,214517° | 5,327929° | 43,214993° | 5,336232° | 570,66 | 863,72 | 293,06 |
| Ile Calseraigne | 43,190199° | 5,381306° | 43,186181° | 5,391561° | 978,91 | 948,00 | -30,91 |
| Ile Jarre | 43,201178° | 5,355180° | 43,193853° | 5,371627° | 1400,32 | 1565,00 | 164,68 |
| Port Pin | 43,204062° | 5,513849° | 43,211172° | 5,521307° | 930,62 | 997,00 | 66,38 |
| **Map of Gulf of Lion** | | | | | | | |
| Ile Riou | 43,179382° | 5,371768° | 43,173686° | 5,398688° | 2075,86 | 2274,00 | 198,14 |
| Ile maire | 43,212390° | 5,328571° | 43,209194° | 5,340005° | 1579,92 | 996,71 | -583,21 |
| Cap Croisette - Cassis | 43,214517° | 5,327929° | 43,214658° | 5,537063° | 16953,95 | 16333,00 | -620,95 |
| Cap Croisette - Cap Creus | 42,319361° | 3,322315° | 43,214993° | 5,336232° | 175533,48 | 192526,00 | 16992,52 |
| Ile Pommegue | 43.276142° | 5.310985° | 43.262178° | 5.287012° | 3621,07 | 2501,00 | -1120,07 |
| Ile Ratonneau | 43.286226° | 5.323944° | 43.280135° | 5.291584° | 4781,82 | 2717,00 | -2064,82 |
| Ile Planer | 43.197204° | 5.228365° | 43.198999° | 5.231435° | 1242,18 | 317,75 | -924,43 |
| Cap Creus - Brescon | 42.319361° | 3.322315° | 43.263251° | 3.501627° | 105101,15 | 105896,00 | 794,85 |


**Table 2: Comparison between the lengths measured in Marsili maps (Gulf of Lion map - Pl. I, page 3 and Cassis map - Pl. II,**
**page 4) and modern map (Google Satellite in QGIS 3.2). The first column listed the names of the 16 sites (the first 8 in Cassis**
**map - Pl. II, page 4 and the second eight in the Gulf du Lion map - Pl. I, page 3). The second and fifth columns indicate the**
**starting and ending points coordinates, the sixth and seventh columns indicate the lengths measured in Marsili and Google**
**Satellite maps, and the last column reports the relative difference between lengths.**

**4.2. Bathymetry: Profiles**
Figure 1 illustrates the transects design from Marsili's maps found in Histoire (Table I) and reconstructed as a bathymetric
profile in Table III of Histoire. Figure 2 showcases a comparative analysis of the depth profiles recorded by Marsili,
converted into meters, alongside those obtained from the EMODnet dataset. One significant observation is that Marsili
underestimates depths greater than 100 meters. This discrepancy likely stems from the limitations of the measurement
techniques available during his era, which impeded accurate depth readings. Additionally, our attempts to georeference
Marsili's maps unveiled more challenges; the spatial alignment of his depth profiles does not correctly match the
geographical features reflected in the EMODnet data. For example, in Marsili's profile F6, a straight line intersection with
both Les Iles and Planier islands reveals an inconsistency, as such a cross-sectional view would not realistically occur in
mapping scenarios like those of EMODnet. Marsili's depiction of seafloor features notably shifts towards the coastline.
This issue is especially apparent in profiles E5 and G7, as shown in Figures 2a-c.

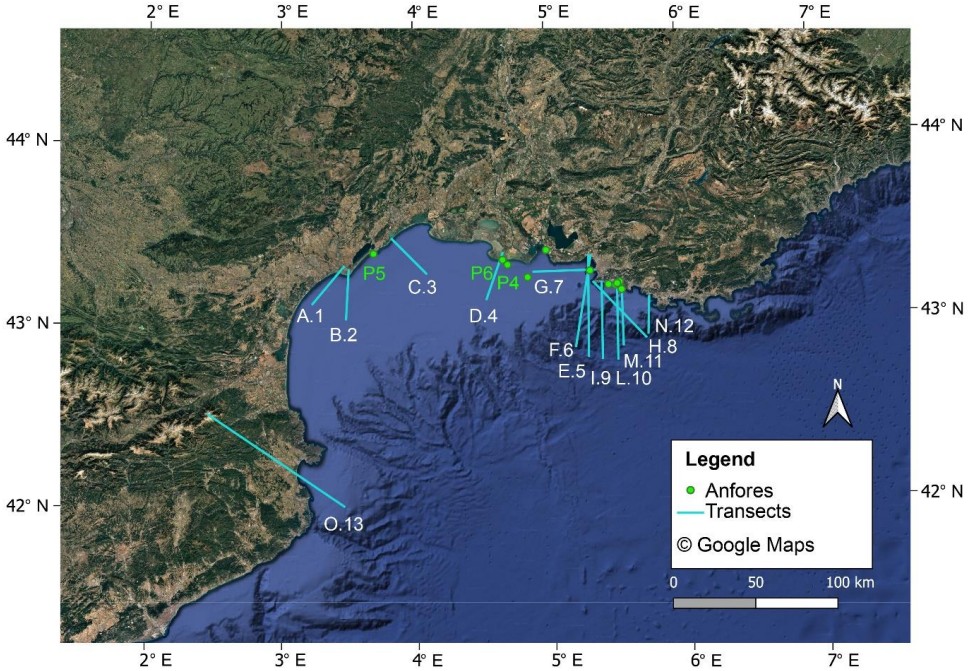


**Figure 1: The map shows the points (P4, P5, P6) in which anfores have been drawn in the Marsili maps on *Histoire* (Table I and II). The anfores represent the sampling points for water weight measurements. Moreover, the figure shows the transects (A1, B2, C3, D4, E5, F6, G7, H8, I9, L10, M11, N12) drawn in the Marsili maps on *Histoire* (Table I). The map has been extrapolated to Google Satellite in QGIS 3.2.**

Despite these differences, Marsili has accurately pinpointed certain geographical features. For example, he clearly outlined the continental shelf in profiles E5 and F6 (see Figures 2a-b) and recognized a canyon in profile H8 (Figure 2d). These correspondences add credibility to Marsili's work, indicating that although his methods might have led to some inaccuracies, he effectively captured essential elements of the seafloor topology that are still pertinent today.

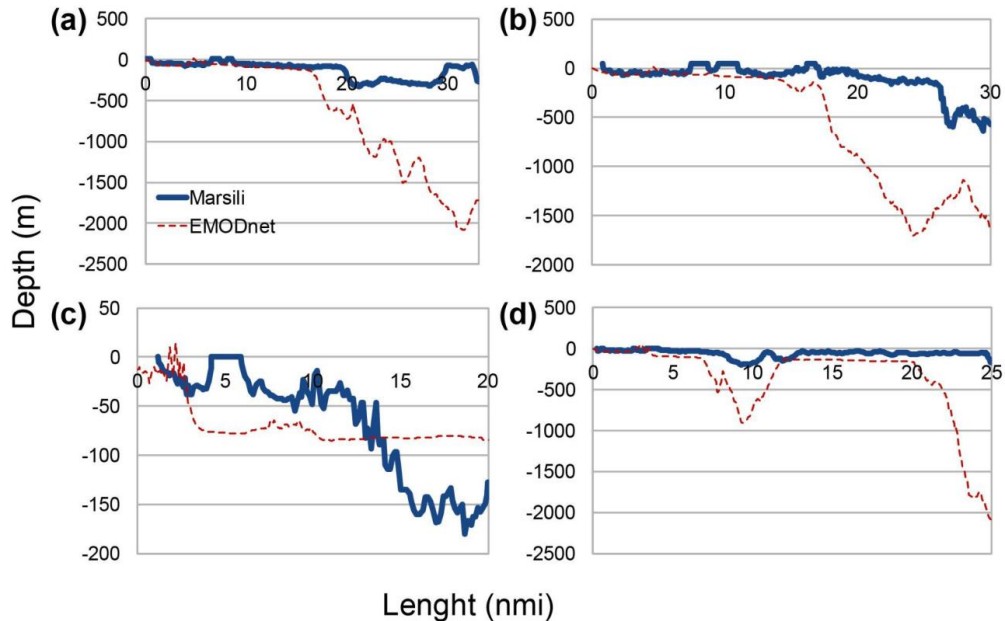


**Figure 2: Comparison between matching EMODnet depth profiles and a) profile E5, b) profile F6, c) profile G7 and d) profile H8 extracted from the figures shown in page 3 and page 4 of Marsili's work.**

### 4.3. Sea Level in Cassis

Applying a bandpass filter to the dataset helps isolate the primary tidal components (M2 and K1) during specific times,
such as the first two weeks of March, revealing oscillations consistent with the M2 tides. The resulting amplitude measures
around 10-12 cm, surpassing the 6.8 cm determined by Woppelmann et al. (2016) in Marseille, which is close to Cassis.
While the values obtained are credible, a more refined methodology and thorough analysis are necessary for accurately
extracting tidal components from the Marsili data.

### 4.4. The water weight measurements

Seawater density is fundamentally tied to the technologies and methods employed in its calculation. One of the earliest
instruments to gauge seawater's 'gravity' or weight was the explorator qualitum. Following this, specific gravity was
evaluated, which is the weight of seawater relative to an equal volume of freshwater. A significant issue arose from the
choice of sample water. Hooke (1635 - 1703) conducted specific gravity measurements using Thames water collected at
low tide in Greenwich (Derham, 1726), while Marsili opted for well water. Phipps became the first to utilize distilled
water during his Voyage toward the North Pole in 1774. Manzella and Novellino (2022) provide descriptions of the
methodologies and technologies for measuring seawater density.
Using a 'non-standard' reference water can introduce errors that must be considered in data analysis. To assess the quality
of the Marsili data, a comparison was made with measurements from recent years. Historical data from the Mediterranean
Sea, spanning from 1806 to 2022, was selected based on proximity to the Marsili measurement points. The TEOS-10
framework was employed to calculate the physical properties of seawater using measurements taken between 1990 and
269  2018.

The historical archive spans the Mediterranean Sea from 1864 to 2022. Initially, the archive was gathered by the former
Italian National Committee for Nuclear Energy (CNEN) through the Center National pour l'Exploitation des Océans
(Cnexo), which later merged with the Institut scientifique et technique des pêches maritimes (ISTPM) to form the current
L'Institut français de recherche pour l'exploitation de la mer (Ifremer). CNEN subsequently provided the initial archive
to the Mediterranean Oceanographic Data Base (MODB) project in the early 1990s. After three years, the MODB data





were incorporated into the Mediterranean Data Archeology and Rescue (MEDAR/MedAtlas) project, which also
integrated additional data. This collective dataset then fed into the SeaDataNet system, where more information was
added. Besides the SeaDataNet entries, the archive includes data from NESDIS/NOAA and public repositories like
SEANOE and PANGAEA, alongside contributions based on personal knowledge. It should be noted that the archive does
not fully cover all positions for the Marsili observations; where there was considerable variability, minimum and
maximum values are provided.
Table 3 presents water density converted from Histoire tables. Data were gathered at various locations in the Gulf of Lion
between June 18, 1706, and January 18, 1707 (illustrated in Figure 1 and Figure 3 with green points). The minimum
possible measurement error has been calculated, yielding results of ±1.23 Kg/m³.

| Latitude [°N] | Longitude[°E] | Site | Data [month day year] | Sea surface water density [Km/m3] | SeaDataNet density (from 1990 to 2018) [Kg/m3] |
|---|---|---|---|---|---|
| 43.277825° | 5.312381° | Marseille islands | June 18th 1706 | 1028,26087 | 1014 - 1028,7 |
| 43.395628° | 4.985985° | Port the Bouc | June 13rd 1706 | 1026,09 | |
| 43.248569° | 4.844061° | Opposite the mouth of the grand Rosne 5 miles offshore | July 14th 1706 | 1021,01 | |
| 43.371753° | 4.600481° | from Chemin du Rosne to Cette in linea retta | July 14th 1706 | 1024,64 | |
| 43.388558° | 3.690230° | Port of Cette | July 15th 1706 | 1028,26 | |
| 43.313049° | 4.655259° | At the mouth of the little Rosne into the sea | October 26th 1706 | 1009,06 | 1014 |
| 43.211923° | 5.533302° | At the mouth of the port of Cassis | December 4th 1706 | 1027,54 | 1027 - 1028,8 |
| 43.199698° | 5.448772° | To the great Chandelle | December 7th 1706 | 1028,99 | |
| 43.211923° | 5.533302° | At the mouth of the Port of Cassis during the storm | | 1028,62 | |
| 43.195922° | 5.500447° | Castello Vieux | January 14th 1707 | 1027,54 | 1027 - 1028,8 |
| 43.184370° | 5.551525° | To Cassidagne | January 18th 1707 | 1030,43 | |
| 43.204045° | 5.513703° | At Port Miou where the river water flows into the sea | January 18th 1707 | 1015,94 | |


**Table 3: The table shows the sampling points coordinates (the anfores in Marsili maps) in which Marsili took water samples to measure water weight. Each coordinate is associated with a site and a date indicated by Marsili in Pl. VII pag. 23. The following columns show the measure of water weight converted in water surface density by Marsili data and the average surface water density extrapolated by SeaDataNET data (from 1990 to 2018).**

Table 3 indicates that Marsili's measurements align well with the latest data. The density value of 1030.43 calculated on
January 18, 1707, is relatively high yet still falls within the range of the estimated minimum error.

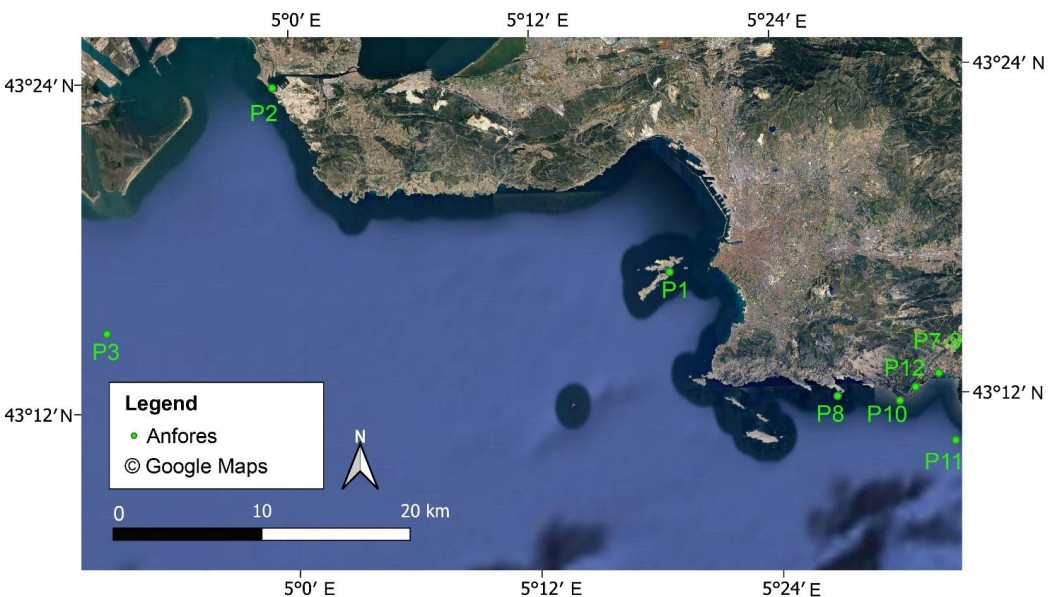


**Figure 3: The map shows the points (P1, P2, P3, P7, P8, P9, P10, P11, P12) in which anfores have been drawn in the Marsili**
**maps on *Histoire* (Table I and II). The anfores represent the sampling points for water weight measurements.**


## 5. Data Availability

The data are available at:
http://oceano.bo.ingv.it/erddap/search/index.html?page=1&itemsPerPage=1000&searchFor=cassis (Locritani et al.,
2024a).
http://oceano.bo.ingv.it/erddap/search/index.html?page=1&itemsPerPage=1000&searchFor=water+denisty+marsili
(Locritani M. & Garvani S., 2024).
http://oceano.bo.ingv.it/erddap/search/index.html?page=1&itemsPerPage=1000&searchFor=marsili (Locritani et al.,
20224b).

## 6. Conclusion

This study presents a comprehensive analysis of eighteenth-century maritime observations in the Gulf of Lion, focusing
on data derived from the significant historical text by Marsili. Collected between 1706 and 1707, this dataset represents
one of the earliest systematic efforts to document the oceanographic characteristics of the region.
While Marsili's data collection methods do not adhere to contemporary standards due to the instrumentation limitations
and the accuracy of units available in his time, we have taken steps to evaluate the relevance of his findings. We aim to
contextualize the results within modern frameworks by applying error margins to the various measurements. For instance,
several parameters, including the weight of surface seawater, were in reasonable agreement with mean values observed
over the last two centuries, suggesting that Marsili's observations hold some validity despite methodological limitations.
Conversely, data related to tidal patterns requires more rigorous methodologies and analyses. The precision of Marsili's
maps was variable, with minimum errors estimated at around 160 meters, indicating potential discrepancies in geographic
representation. These inaccuracies highlight the challenges faced by early oceanographers, yet they also underscore the
pioneering nature of Marsili's work.





Despite these limitations, Marsili's contributions remain invaluable. His work not only provided the first systematic
description of the Gulf of Lion's continental shelf and abyssal regions but also laid the groundwork for future
oceanographic studies in the area. By bridging the historical insights and contemporary analysis, this study reaffirms the
significance of early oceanographic research and its relevance to our understanding of marine environments today.
Through this exploration, we honour Marsili's legacy while acknowledging the evolution of oceanographic methodologies
over the centuries.

**Author contribution:** ML conceptualized and administrated the research, validated the georeferenced maps and wrote
the original draft preparation, SG investigated the historical part of the project and wrote original draft preparation, GM
validated sea level data and supervised the entire study and wrote review and editing of the text, GT validated the
bathymetric data and wrote original draft preparation, AG providing funding acquisition as MACMAP project
coordinator.
**Competing interests:** Author GM is a member of the editorial board of the journal.
**Publisher's note:** Copernicus Publications remains neutral with regard to jurisdictional claims in published maps and
institutional affiliations.
**Acknowledgements:** The authors are deeply indebted to the owner of the volume Histoire physique de la mer (Marsili,
1725) allowed to study on the original book. The authors are grateful to Grammarly and CatGPT who helped correct the
grammar of the entire paper and rephrase the sentences in the conclusion section.
**Financial support:** This research has been developed in the framework of the MACMAP (A Multidisciplinary Analysis
of Climate change indicators in the Mediterranean And Polar regions) project funded by Istituto Nazionale di Geofisica
e Vulcanologia (Environment Department), (https://progetti.ingv.it/index.php/it/progetti-
dipartimentali/ambiente/macmap).

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

Appendix A

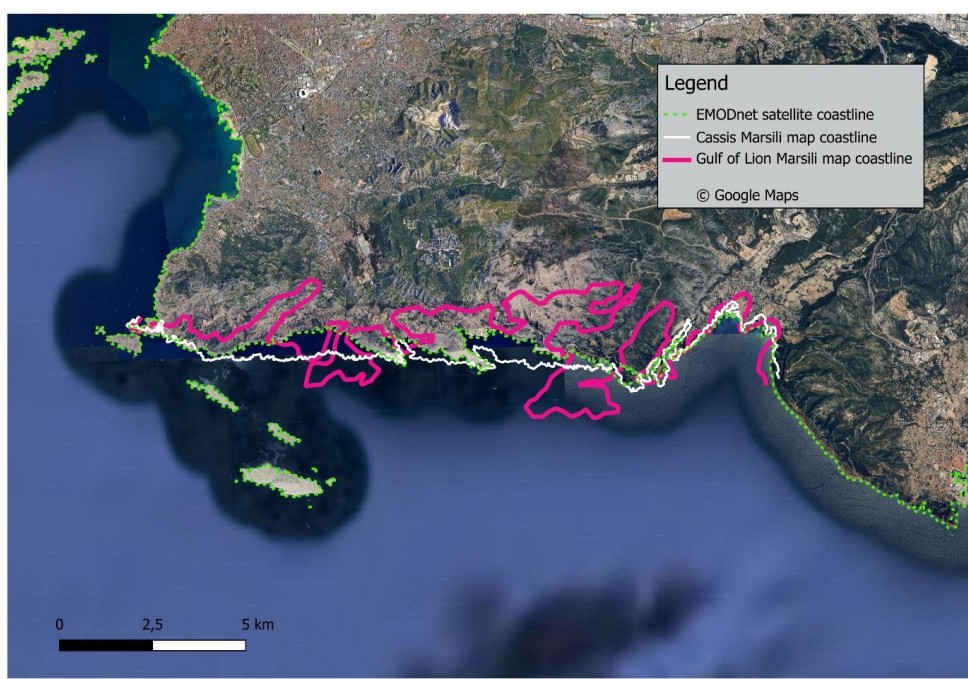


**Figure A1: Comparison of three coastlines in the area near Cassis. The dotted green one is the coastline extrapolated from**
**EMODnet data, the purple line represents the digitised coastline derived from the historical map of the Gulf of Lion. In**
**contrast, the white line depicts the coastline extrapolated from the historical map of Cassis. The map has been extrapolated to**
**Google Satellite in QGIS 3.2.**
Appendix B
**Marsili biography and the scientific contest of reference**
Luigi Ferdinando Marsigli (Bologna, 1658-1730) was a natural scientist and soldier whose remarkable and idiosyncratic
accomplishments in both fields warrant recognition. During his youth, Marsigli accompanied his father and other family
members on various journeys, which allowed him to become acquainted with numerous distinguished scholars, including
the renowned librarian Antonio Magliabechi (1633-1714), the medical philosopher Tomaso Cornelio (1614-1684), and
Giovanni Borelli (1608-1679). These formative experiences fostered in Marsigli a profound and enduring interest in the



marvels of the natural world. Marsigli pursued extensive studies, albeit with irregular intervals. At the University of
Bologna, he studied under the anatomist Marcello Malpighi (1628-1694), the botanist Lelio Trionfetti (1647-1722), and
the mathematician and astronomer Geminiano Montanari (1632-1687), with whom he maintained a regular
correspondence. As a result of the influence of his teachers, Marsili was able to embrace the contemporary principles,
methodologies and approaches of the Royal Society of London. In particular, Marsigli was profoundly influenced by the
instructions on experimentation and instrumentation that he received from his mentor, Geminiano Montanari. In 1665,
Montanari established the Accademia della Traccia or dei Filosofi in Bologna. This institution combined mathematical
and experimental approaches and emphasised the instrumentation, discoveries, and experiments conducted by the Royal
Society. Marsili's expertise was also significantly shaped by one of the founders of the Royal Society. Robert Boyle
(1627-1691) operationalised Marsili's procedure by translating the intentions expressed by Boyle in his treatises (*Tracts*
*consisting of Observations about the Saltness of the Sea,* London,1674), this involved adapting the procedure to the
specific context.
Marsigli's first book of 1681, *Osservazioni intorno al Bosforo Tracio*, dedicated to Queen Christina of Sweden, reported
observations and experiments he had carried out during and shortly after his year-long visit to Istanbul in 1679 and 1680.
While this work is recognised by oceanographers as the first description of a simple laboratory-scale hydrological
experiment to show how differences in the density of two water masses lead to two-layer currents, another treatise by
Marsigli consecrated him as the lost father of oceanography (Olson,1958): *Histoire physique de la mer*, published in 1725.
This remarkable treatise was written during the time Marsili spent in France. After a few years of training in chemistry in
Paris (1704-1706), Marsigli expressed his desire to study the organic structure of the Earth. He first travelled to
Montpellier, where he was accepted as a member of the newly formed 'Societe royale', modelled on the Paris Academy
of Science (Carpine-Lancre and McConnell, 1985), during which time he contributed by providing a list of valuable
correspondents and suggesting essential books and journals for the Society to acquire. He illustrated the members of his
research with watercolour illustrations, including a location map, a sectional drawing of the seabed showing where the
coral was dredged and the apparatus used. At first, Marsigli thought it was a mineral concretion like those he had seen in
petrified springs. The true nature of coral - mineral, plant or animal - was debated from antiquity until the late 18th
century. After Montpellier, Marsigli moved on to Cassis, where he regularly joined the coral fishermen, taking
temperature readings, making soundings and collecting water samples. Marsigli observed the tides, studied waves and
currents and documented the colours of the water. In his laboratory on land, he measured the gravity of seawater samples,
analysed corals and used a microscope to study their structure and function.
In a letter to Abbé M. Bignon, he outlined his research intentions: "I have begun research on the history of the sea, where
I hope to treat the nature of the water of the sea and its diverse movements; of the differences of the bottoms of the sea,
which seem to me to be related to the structure of the mountains, of the effect of winds on this water, of the nature of fish
developed through analysis of the vegetation growing on the bottom of the sea."
This productive period, spanning a mere two years, was interrupted by Marsigli's second call to military service, this time
by Pope Clement XI. In 1708, Marsigli briefly departed for the Adriatic coast to command the Papal States' troops, seizing
the opportunity to conduct limited oceanographic studies in those waters. Upon his return to Cassis, he commenced work
on his treatise, publishing a brief extract in 1711: *Brieve ristretto del Saggio fisico*. In 1715, he was once more summoned
to serve the Pope. This was to be his final deployment. Upon the cessation of hostilities, he relinquished his military
obligations and subsequently dedicated the remainder of his life to the pursuit of his studies. His extensive collection of
scientific materials was donated to his hometown of Bologna on 13 March 1714, which marked the establishment of the
Bologna Institute of Science and Art. In 1715, he was elected to the Paris Academy of Sciences. He subsequently travelled
to England, where he formed a friendship with Newton and Halley. In 1722, he was elected a Fellow of the Royal Society.
The results of his two-year research at Cassis were published in 1725 in the form of his monumental work, *Histoire*
*Physique de la Mer*, which he dedicated to the Academy of Sciences in Paris. The work was eventually published in
Amsterdam in 1725.
Areometer or hydrostatic ampulla Note:
The instrument employed for the measurement of weight in both surface and deep water is designated as a hydrostatic
ampulla, hydrostatic bulb, or areometer. The description of an instrument called a "hydrostatic ampulla," constructed
following Montanari's technical instructions and closely resembling the apparatus utilized by Marsigli in the Sea of
Provence, can be found in a posthumous publication by Montanari in the cultural journal "La Galleria di Minerva" in
1704 (Montanari, 1696). The water weight instrument, which consisted of a small, long-necked bottle weighted with
sufficient lead shot to enable it to float upright, was used by Marsili in the Bosphorus. This instrument was purchased by
Montanari himself (Soffientino & Pilson, 2005). It is noteworthy that Marsigli himself stated regarding the use of the
hydrostatic bulb that: "*l'altra parte che compone la natura di queste acque è il sapore salso, universale a tutti i mari che*
*più, e meno sono di esso abbondanti, come l'esperienza me l'ha mostrato mediante il peso rilevato con l'Ampolla*





*Idrostatica, secondo gli insegnamenti datimi dal Sig. Dott. Montanari, celebre matematico, e mio riverito Maestro, che*
*fin da primi anni cominciò a dimostrarmi i principi di simili studi, e tra gli altri, il fondamento dell'uso di tale Istromento,*
*e la perfezzione alla quale col suo nobile ingegno l'ha ridotto; e ne conservo una lettera in cui se ne parla diffusamente*"
(Marsili, 1681, pag. 71). In the *Histoire Physique de la mer*, the areometer is illustrated in Plate VII, accompanied by its
weight in air (1 ounce, 3 drachms and 10 grains) and the lead rings with the corresponding weights. The ampoule is used
to measure density, which was also referred to as 'gravity'. When Marsili measures a weight, he is measuring gravity.
From the concept of generic gravity, we move on to that of specific gravity, whereby the measurement is related to a
sample. This is a concept that emerged from the French Revolution onwards and is now used for water. However, Marsili
uses rainwater instead.
**Additional references**
Adamson, G.C.D., Private diaries as information sources in climate research, Wiley Interdiscip. Rev. Clim. Change, 6,
599-611 pp., 10.1002/wcc.365, 2015.
Adamson, G. C. D., Hannaford, M. J., & Rohland, E. J., Re-thinking the present: the role of a historical focus in climate
change adaptation research. Global Environmental Change, 48, 195-205 pp., 2018.
Allan, R., Endfield, G., Damodaran, V., Adamson, G., Hannaford, M., Carroll, F., ... & Bliuc, A. M., Toward, integrated
historical climate research: the example of Atmospheric Circulation Reconstructions over the Earth. Wiley
Interdisciplinary Reviews: Climate Change, 7(2), 164-174, 10.1002/wcc.408, 2016.
Camuffo, D., Sturaro, G., Sixty-cm Submersion of Venice Discovered Thanks to Canaletto's Paintings,. Climatic Change
58, 333–343 pp., https://doi.org/10.1023/A:1023902120717, 2003.
Camuffo, D., Sturaro, G., Use of proxy-documentary and instrumental data to assess the risk factors leading to sea
flooding in Venice, Global and Planetary Change, Volume 40, Issues 1–2, 93-103 pp., ISSN 0921-8181,
https://doi.org/10.1016/S0921-8181(03)00100-0, 2004.
Camuffo, D., The Treatise on Waters by Cornaro (1560) and a quantitative assessment of the historical sea surges "Acqua
Alta" in Venice. Climatic Change 176, 18, https://doi.org/10.1007/s10584-023-03492-6, 2023.
Della Valle, A., Camuffo, D., Becherini, F. et al. Recovering, correcting, and reconstructing precipitation data affected
by gaps and irregular readings: The Padua series from 1812 to 1864. Climatic Change 176, 9,
https://doi.org/10.1007/s10584-023-03485-5, 2023.
Folland, C. K., & Parker, D. E., Correction of instrumental biases in historical sea surface temperature data. Quarterly
Journal of the Royal Meteorological Society, 121(522), 319-367 pp., 1995.
MacEachren, A. M., The evolution of thematic cartography/a research methodology and historical review. The Canadian
Cartographer, 16(1), 17-33 pp., 1979.
Pfister, C., Brázdil, R., Glaser, R., Barriendos, M., Camuffo, D., Deutsch, M., & Rodrigo, F. S., Documentary evidence
on climate in sixteenth-century Europe. Climatic change, 43, 55-110 pp., 1999.
Wilkinson, C., Woodruff, S. D., Brohan, P., Claesson, S., Freeman, E., Koek, F., & Wheeler, D.. Recovery of logbooks
and international marine data: the RECLAIM project. International Journal of Climatology, 31(7), 968-979 pp., 2011.