# Peer review of "A revisiting of early 18th century environmental data to identify Gulf of Lion properties before the industrial era"

_Earth System Science Data, 2025_

## Referee Comment (RC1)

This is an interesting paper reporting on the oceanographic observations made in the gulf of Lion in the Mediterranean sea by Luigi Ferdinando Marsili, author of one of the first methodical books describing observations of sea properties, his celebrated "Histoire Physique de la Mer", 1725.

The greatest value of this paper is not, in my judgement, the "reconstruction" in modern units of the measurements made by Marsili in the gulf and the description of its properties three centuries ago. The reliability of this "reconstruction" is not only doubtful considering the empiricism of the original measurements; but is also of very limited scientific interest. What is interesting is the figure and personality of Marsili himself and his "baconian" approach to science and investigation of sea properties. For me the major value of the paper is "historical" and the most interesting section is Marsili's biography, Appendix B.

Looking also at the bibliography, one notices the numerous but "scattered" papers stemmed by examining different aspects and parts of the Histoire. Therefore I make a suggestion : to assemble all these different contributions into a unique, comprehensive article, maybe a special issue of Oceanography, fully dedicated to L.F. Marsili.

Exactly in the spirit of honoring "Marsili's legacy" as stated in line 321 of the paper.

Paola Malanotte

---

## Author Response (AR1)

Referee 1 (Prof Paola Malanotte Rizzoli)

This is an interesting paper reporting on the oceanographic observations made in the gulf of Lion in the Mediterranean sea by Luigi Ferdinando Marsili, author of one of the first methodical books describing observations of sea proper es, his celebrated "Histoire Physique de la Mer", 1725. The greatest value of this paper is not, in my judgement, the "reconstruction" in modern units of the measurements made by Marsili in the gulf and the description of its proper es three centuries ago. The reliability of this "reconstruction" is not only doubtful considering the empiricism of the original measurements; but is also of very limited scientific interest. What is interesting is the figure and personality of Marsili himself and his "baconian" approach to science and investigation of sea properties. For me the major value of the paper is "historical" and the most interesting section is Marsili's biography, Appendix B. Looking also at the bibliography, one notices the numerous but "scattered" papers stemmed by examining different aspects and parts of the Histoire. Therefore I make a suggestion: to assemble all these different contributions into a unique, comprehensive article, maybe a special issue of Oceanography, fully dedicated to L.F. Marsili. Exactly in the spirit of honoring "Marsili's legacy" as stated in line 321 of the paper.

*The authors express their gratitude for the suggestions and acknowledge the historical significance of the work. This article forms part of a historical dissertation (PhD Thesis) that is nearing completion. The thesis explores the marine research conducted by L. F. Marsili in the Gulf of Lion between 1706 and 1707, which laid the groundwork for Histoire Physique de la Mer (1725). The study examines his methodology, the instruments he utilised, and the impact of the Royal Society and his Italian mentors. The historical context of his discoveries is also analysed, including his mistaken identification of coral polyps as flowers.*

*The evolution of Marsili's work is then traced, from his initial study on the connection between mountains and the sea, to the publication of the Histoire. The re-editing and summarising of the arguments of the thesis leads to the consideration of the idea of writing a historical paper to honour Marsili's legacy, whilst acknowledging the evolution of oceanographic methodologies over the centuries.*

*The authors also emphasise the necessity of providing relevance and traceability to the data collected over 300 years ago by Marsili using rudimentary instruments and methodologies. The data contained in the Marsili book and re-analysed in this paper could be used for the study of environmental changes (natural/man-induced), with the awareness of the error that occurs in the measurements, or to compare with other historical data taken in other places. This underlined sentence has been introduced at the end of the Introduction to better explain the purpose of this paper (lines 70 – 72)*

Referee 2:

This is one of the important papers for historical oceanography and it focuses in the reconstruction of environmental measurements translating them in modern units from literature records. I recommend the publication subject to minor revisions mainly adding better images, more discussion and interpretation of the results, in addition to some nomenclature issues which I suggest should be corrected by the authors before publication.

*The authors thank the referee for having very positively welcomed the proposal contained in this paper and explained on the basis of the comments of referee 1. They also thank her/him for the contribution*

*given for the improvement of the paper.*

I have some major remarks:

1. I believe it would be nice to have the Marsili's maps reproduced (Gulf of Lion map - Pl. I, page 3 and Cassis map - Pl. II, page 4) because the discussion is otherwise very difficult to follow. There is somehow a little confusion between "distance" along the transects and "lengths" discussed. I suggest that instead of "lengths", "distance along transects" is used.

   *Maps have been included as Figure 1 and Figure 2 and 'length' has been used instead of 'distance'*

2. Section 4.1 should not be called "Bathymetry: historical map analysis" but "Measurement sections: historical map analysis". *done - thanks for this correction.* Furthermore several times there is "map" instead of "distance". In oceanography we do not use the word map to indicate distance on the earth.

   *Map(s) is used very often and could lead to confusion. In some cases we replaced 'map' with 'bathymetry chart' (line 106), in other cases with 'distance' (lines 169, 267), and left 'map' when a diagrammatic representation of physical features was intented.*

3. Furthermore why calculate the "mean" distance? The text is reported here: [The mean distances are 2912 ± 6057 m for the Gulf du Lion map and 324 ± 472m for the Cassis map.] What is the oceanographic significance of a mean "section distance"? I suggest to eliminate this. *- Deleted*

4. Furthermore I believe the authors should conclude in a more precise manner that to have distances matching the present ones you need to consider a different coastline, as shown in Appendix A. This is an important consideration which is not presented adequately.

   *This is a very important contribution given by the referee. We added this sentence in the conclusions: 'In this article and in Appendix A it is demonstrated that to have a good matching between the distances calculated from the Marsili maps and the real ones it is necessary to consider a correct coastline, which was one of the major efforts made for this paper' (lines 340 – 342)*

5. Your statement: The minimum possible measurement error has been calculated, yielding results of ±1.23 Kg/m$^3$. Please define how you estimated errors, this is important.

   *Explanation of estimated error is given in paragraph 4.4 lines 308 – 310.*

   Furthermore, why showing Fig. 3 if the amphores were shown already in Fig. 1?

   *In our opinion, both maps are needed because they are at two different scales.,* con una sola mappa non si vedono tutti i punti, abbiamo risposto QUALE LINEA?

6. Throughout the text, even in the conclusions, it is not mentioned that Marsili did his first measurements of density in the Bosphorus, much before the ones in the Gulf of Lion. In addition he used a different reference water, this is in my opinion an important methodological aspect. Please insert somewhere this relevant note and refer to Pinardi et al. (2018) for the discussion on the other reference waters.

   Results and discussions are including the requested information (lines 295 – 309;

Detailed comments

1. Please give a DOI for the EMODnet_satellite_coastline_MSL, it cannot be referenced in this way;

https://doi.org/10.12770/cf51df64-56f9-4a99-b1aa-36b8d7b743a1

2. Line 215, the formula was first presented by Pinardi et al. (2018) and it should be referenced,

*Reference was added*

3. From line 270-280: I suggest you do not talk about the history of past projects/efforts to collect the data in the SeaDataNet archive but you just reference the database which has a DOI. This is not a paper on the recent historical database and it should be enough to give its referencing in modern literature.

*This part has been updated deleting some sentences and adding DOIs*

*The authors have made the following changes based on the reviewers' suggestions:*

Line 6: Changed the authors order: Marina Locritani[1], Sara Garvani[1,2], Giancarlo Tamburello[1], Antonio Guarnieri[4], Giuseppe Manzella[3]

Line 9 and 11: changed order of affiliations

Line 14: Histoire Physique de la Mer made in italic format

Line 42 De fundo Maris made in italic format

Line 43, 44, 59, 134: Histoire made in italic format

Line 60 Brieve Ristretto made in italic format

Line 61-62: Replaced the sentence: "This is not the first occasion to analyze Marsili's measurements compared to contemporary data". With: "This is not the first time that Marsili's measurements have been analyzed by comparing them with contemporary data."

Line 70,71,72 added the sentence: The data contained in the Marsili book and re-analysed in this paper could be used for the study of environmental changes (natural/man-induced), with the awareness of the error that occurs in the measurements, or to compare with other historical data taken in other places.

Line 106: replaced "map" with "bathymetric chart"

Line 109: Added "Figure 1"

Line 111: Added "Figure 2"

Line 113: Delated sentence "Neither map features a Coordinate Reference System and they are based on earlier maps"

Line 125 and 126: Added Figure 1 and caption: "Maps included in Histoire Physique de la Mer in Carte du Golfe del Lion entre le Cap Sisie en Provence et le Cap de Quiers en Roussillon in Table I, page 3."

Line 128 and 129: Added Figure 2 and caption: Maps included in *Histoire Physique de la Mer* in *Carte Particuliere de la Coste*, Table II, page 4.

Line 168: Replaced "these maps" with "them"

Line 169: Replaced "maps" with "distances"

Line 173, 175, 191, 225, 228: Replaced "distances" with "lengths"

Line 191: Added "https://doi.org/10.12770/cf51df64-56f9-4a99-b1aa-36b8d7b743a1"

Line 186: Replaced "vertical distance" with "depths"

Line 216: Added reference "(Pinardi et al., 2018)"

From Line 218 to 219: Added sentence "taken from Plate VIII page 23 of the Histoire Physique de la mer that shows the weights of distilled surface water, which is used by Marsili as the reference water (1 ounce, 3 drachmas, 30 grains equivalent to 1000 $Kg/m^3$), see Table A1 in Appendix

Line 220: Delated sentence "from Marsili's measurements"
Line 224: Replaced title: "Bathymetry: historical map analysis" with "Measurement sections: historical

map analysis"

Line 226: Replaced "each map" with "distance"
Line 230: Added "lengths measured" two times

Line 231: Delated "The mean lengths distances are 2912 ± 6057 m for the Gulf du Lion map and 324 ±

472 m for the Cassis map."

Line 232: Added "as highlighted by Table 2."

Line 232: Delated "evidenced by the standard deviations exceeding the mean values."

Line 243: Changed 4.1 with 4.2

Line 244: Replaced Figure 1 with Figure 3

Line 245: Replaced Figure 2 with Figure 4

Line 253: Replaced Figure 2 with Figure 4

Line 255: Replaced Figure 1 with Figure 3

Line 260: Replaced Figure 2 with Figure 4

Line 261: Replaced Figure 2 with Figure 4

Line 264: Replaced Figure 2 with Figure 4

From line 285 to 291: Delate the sentence "Initially, the archive was gathered by the former Italian

National Committee for Nuclear Energy (CNEN) through the Center National pour l'Exploitation des

Océans (Cnexo), which later merged with the Institut scientifique et technique des pêches maritimes (ISTPM) to form the current L'Institut français de recherche pour l'exploitation de la mer (Ifremer). CNEN subsequently provided the initial archive to the Mediterranean Oceanographic Data Base (MODB) project in the early 1990s. After three years, the MODB data were incorporated into the Mediterranean Data Archeology and Rescue (MEDAR/MedAtlas) project, which also integrated additional data."

Line 291 and 292: delated the sentence "then fed into the SeaDataNet system, where more information was added. Besides the SeaDataNet entries, the archive"

Line 392 and 293: Added "(DOI 10.12770/2a2aa0c5-4054-4a62-a18b-3835b304fe64)"

Line 296: Replaced "tables" with "Histoire (Pl. VII page 23 Table I)"

Line 297: Replaced Figure 1 with Figure 3 and Replaced Figure 3 with Figure 5

From line 308 to 310: Added the sentence "For example Marsili measured 1 ounce 3 drachmas 28 grains equivalent to 997,10 $Kg/m^3$ to the Montpellier fountain of St. Giles. This value is surely wrong, calculating the difference between 1000 $Kg/m^3$ and the mean eighth wrong value (998,77 $Kg/m^3$) present in the Pl. VII Table 2, the result is ±1,23 $Kg/m^3$."

Line 320: Replaced Figure 3 with Figure 5

From line 340 to 342: Added the sentence: "In this article and in Appendix A it is demonstrated that to have a good matching between the distances calculated from the Marsili maps and the real ones it is necessary to consider a correct coastline, which was one of the major efforts made for this paper".

Line 329: Replaced Figure 3 with Figure 5.

Line 446 in appendix: Added the table A1 and caption "Table A1: Reconstructed water density measured from surface water, distilled surface water and water cistern, fountains and wells calculated from the data included in Pl. VIII page 23 of Histoire."

Line 449 in appendix: Added the table A2 and the caption "Table A2: Reconstructed water density measured from water rivers, fountains and wells calculated from the data included in Pl. VII page 23 of *Histoire*."